# Risk Factors of Lyme Disease: An Intersection of Environmental Ecology and Systems Science

**DOI:** 10.3390/healthcare7020066

**Published:** 2019-04-30

**Authors:** Nasser Sharareh, Rachael P. Behler, Amanda B. Roome, Julian Shepherd, Ralph M. Garruto, Nasim S. Sabounchi

**Affiliations:** 1Department of Population Health Sciences, University of Utah, Salt Lake City, UT 84108, USA; 2Department of Chemistry, the State University of New York at Binghamton, Vestal, NY 13902, USA; rbehler1@binghamton.edu; 3Department of Anthropology, the State University of New York at Binghamton, Vestal, NY 13902, USA; aroome1@binghamton.edu (A.B.R.); rgarruto@binghamton.edu (R.M.G.); 4Department of Biological Sciences, the State University of New York at Binghamton, Vestal, NY 13902, USA; jshepher@binghamton.edu; 5Department of Systems Science and Industrial Engineering, the State University of New York at Binghamton, Vestal, NY 13902, USA; sabounchi@binghamton.edu

**Keywords:** ticks, human behavior, regression, urban planning, simulation modeling, rodents, vegetation

## Abstract

Lyme disease (LD) cases have been on the rise throughout the United States, costing the healthcare system up to $1.3 billion per year, and making LD one of the greatest threats to public health. Factors influencing the number of LD cases range from environmental to system-level variables, but little is known about the influence of vegetation (canopy, understory, and ground cover) and human behavioral risk on LD cases and exposure to infected ticks. We determined the influence of various risk factors on the risk of exposure to infected ticks on 22 different walkways using multinomial logistic regression. The model classifies the walkways into high-risk and low-risk categories with 90% accuracy, in which the understory, human risk, and number of rodents are significant indicators. These factors should be managed to control the risk of transmission of LD to humans.

## 1. Introduction

Lyme disease (LD) has become the most common tick-borne infection in the northern hemisphere, with cases on the rise in parts of the United States, Europe, and Asia [1,2,3,4]. Within the United States alone, more than 300,000 cases are estimated to occur annually, nearly tripling over the past two decades, with the highest incidence rates found in the northeast and upper Midwest [5,6]. Due to possible infection at nymphal and adult tick life cycle stages and the potential for long-term complications, LD has become a major public health concern with associated economic effects, costing the United States healthcare system up to $1.3 billion annually [7,8,9]. The infection is transmitted through the bite of the *Ixodes scapularis,* the black-legged tick, harboring *Borrelia burgdorferi,* which is the causative agent of Lyme disease [10]. The agent is found in different reservoir hosts, mostly notably the white-footed mouse (*Peromyscus*
*leucopus*), which is the most competent reservoir host in the northeastern United States [11]. Initial symptoms of infection are often characterized by rash, severe fatigue, joint pain, and neurological symptoms [12]. If left untreated, symptoms may progress and present as complex multi-system manifestations [12,13].

During the past decade, LD cases have been reported in regions spanning an unprecedented geographic range including the northeastern and mid-Atlantic states as well as the upper Midwest and Canada [1,14,15,16,17,18,19]. In the northeastern United States alone, there has been a 320% increase in the number of high-incidence counties [14]. This northern spread of LD has been correlated to dynamic abiotic and biotic changes related to global temperature changes, landscape features, land use, and levels of biodiversity, amongst others, which impact vector and reservoir populations [18,19,20]. These variables may alter human risk throughout the tick life cycle and include other environmental risk factors, which may increase the risk of LD [21,22,23].

Predictive studies investigating the northward migration of *I. scapularis* have been able to link tick expansion with climate change [23]. This model suggests that *I. scapularis* habitat will increase approximately 213% northward by 2080 (with a subsequent retraction in southernmost borders), which may help identify regions of high LD risk throughout this period. Brownstein et al. [23] suggested the incorporation of vegetation data in models to accurately predict future tick distribution at a higher resolution and the importance of identifying ecological and climatic factors that correlate with high LD transmission risk, especially in highly endemic regions. Other studies have assessed the risk of LD transmission (either through LD incidence or relative tick density) with various ecological factors, including vegetation type, habitat fragmentation, temperature, and humidity [23,24,25,26]. However, few studies have been able to correlate multiple ecological risk factors and comprehensive information on the spatial/ecological dynamics of LD, as LD transmission to humans requires close spatial proximity between human hosts and infected ticks. Such studies may be useful for public health policymakers to address this growing public health issue. The most viable mitigation strategy may be via public awareness, and subsequent intervention and ecological modifications [27].

Recently, McClure and Diuk-Wasser [28] developed a model that would help distinguish between entomological risk and LD incidence risk, as the two may not necessarily yield the same correlation due to the scale at which risk is assessed versus the scale at which incidence occurs [28]. Using this model, LD incidence risk was assessed in relation to various ecological characteristics including forest cover, size, isolation, and connectedness of small-scale patches across a 12-country region in New York state. Simulated landscapes were used to translate risk from a small patch scale to that which would encompass a larger-scale landscape [28]. It was determined that forest patch size and entomological risk were negatively correlated, which is consistent with LD incidence distribution in New York state as well as those predicted by other current fragmentation models [28].

Ecological variables including cover type, patch characteristics, and environmental stability are among the current metrics that are being used to correlate LD incidence, risk, and tick density using simulation models, various algorithms, and regression methods. In a recent study evaluating LD risk in southeastern Canada [29], environmental risk algorithms were used to evaluate the effect of several ecological variables on potential LD risk in the region. Formulations of the used environmental stability algorithms included those relating to temperature, vegetation, and rainfall. Algorithm performance was assessed through receiver operating characteristics and area under the curve methods [29]. From this study, high-risk geographic locations were identified that could be correlated to LD incidence to better assess risk.

Additional variables such as litter depth, humidity, and distance from trafficked forest trails were investigated in a risk assessment model developed by Ripoche et al. [30] for the southern region of Quebec, Canada. This study focused on investigating smaller-scale habitat variables that may be more applicable to local prevention approaches. Variables specific to selected plots such as canopy cover, forest floor vegetation cover, tree and shrub density, and wetland prevalence were correlated to infected nymph and adult tick abundance in an emerging LD region using regression models. It was determined that variation in nymph abundance and clustered distribution patterns resembled studies performed on other emerging LD areas within the northeastern United States. Additionally from this study, forest nymph density was found to be negatively correlated with distance from trails [30]. This information points to key information regarding risk variation that may also have public policy applications. Similarly, Jackson et al. [31] investigated the correlation between ecological landscape features and LD cases within a 12-county region in Maryland. This study determined that forest edges bordering herbaceous vegetation, as well as half-forested land, are correlated with the highest LD risk, providing further information on landscape modification tools for potential use in public policy.

Although ecological risk factors are an important consideration in evaluating LD risk, human interaction with the ecosystem and correlation to the number of LD cases may also introduce additional insight. In a study published by Seukep et al. [16], environmental characteristics coupled with demographic information were studied for the correlation with LD contraction risk in Virginia, where the incidence of LD is increasing. Using Geographic Information System (GIS) and a spatial Poisson regression model, different types of vegetation and human demographics were correlated to LD risk over a period of five years [16]. Vegetative data was recovered from the National Land Cover Database and LD case data from the Virginia Department of Health. This study determined that several ecological and demographic variables were positively correlated with LD incidence, including the percentage of developed forest and herbaceous segments within a forested area, in addition to contrast-weighted edge density and the total edge contrast index, among other factors (27). The percentage of herbaceous cover was positively correlated with LD incidence rates, as well as the mixture of herbaceous and forest lands. Forest cover percentage was negatively correlated with LD risk, which disagrees with the results of a similar study conducted in Maryland [31]. This may be due to geographical differences in forest types between the two regions [16].

Our current study is an effort to evaluate the LD risk and the risk of exposure to infected ticks associated with vegetation, walkway types, human risk factors, and vector and reservoir data on a mid-sized university campus in upstate New York, where LD is endemic. We anticipate that information gleaned from this study may be useful in the formulation of public health policy strategies to address LD risk. In addition, it will help urban planners identify cover types for different regions.

## 2. Data

Our data for modeling comes from the built environment of a mid-sized university campus in the northeastern United States (US) that is endemic for Lyme disease. We define built environments according to the criteria of Srinivasan et al. [32]: places where people live, work, and spend their leisure time, such as parks, school campuses, and other human-made or altered external environmental space where people are regularly perambulating or congregating.

The data includes four different modules. Vegetation data were gathered specifically for the purpose of this research article from 22 different walkways at the mid-sized university campus mentioned above; data for other modules were obtained from a previous published article [33] available for all those 22 walkways. Sharareh et al. [33] used systems science techniques including data analysis and simulation to develop a model in order to propose policies that are capable of reducing LD cases. Below, we explain each module in detail.

### 2.1. Vegetation Data

Percent canopy cover (>5 m), understory (1–5 m), and ground cover (<1 m) were estimated visually during June to August 2014; 10 m on each side of each walkway were averaged along the length of the walkway. Table 1 shows the percentage of coverage within each level of vegetation. The total of these percentages may be greater than 100%, because each one represents the percentage of area that was covered with that specific type of cover, and each type might overlap in some areas. In the built environment living areas of Hillside and Susquehanna Walkways (see Figure 1), the canopy (though sparse) was largely white ash, Norway maple, and Norway spruce. The understory was dominated by autumn olive, honeysuckle, and multiflora rose, and the ground cover was dominated by grasses (some lawn), goldenrods, and herbaceous legumes. Along the Nature Preserve walkways, the canopy was dominated by white ash, red maple, sugar maple, and gray birch. The understory in the Nature Preserve (see Figure 1) was dominated by autumn olive, honeysuckle, and silky dogwood in open areas, and by witch hazel where the canopy was near complete. The ground cover was goldenrods and grasses in the open, and ferns and white grass under the canopy.

We used GIS to create the walkways on the map for illustration purposes, as shown in Figure 1.

### 2.2. Tick-Rodent Data

Black-legged ticks (Ixodes scapularis), the primary vector, were collected within 3 m on each side of all 22 walkways for a total area of 71 534 m^2^ (7.1 ha) during two consecutive years, 2013 and 2014. Ticks were collected using the traditional dragging method except for winter months. Walkways were surveyed in the fall and spring, and each one was surveyed between one and four times. Ticks were transported from the cloths to the laboratory for pathogen analysis. Tick densities could be calculated using the number of ticks collected and the area dragged, which is discussed in detail by Sharareh et al. [33]. However, for two paths—HS6 and SQ1—no data was gathered in 2014, and we used 2013 data for both 2013 and 2014. Table 1 shows the density of nymphal ticks, the density of adult ticks, and the density of infected ticks (densities are per 1000 square meters).

In addition, we collected mice throughout the campus, although this data was very limited; hence, the population of mice was derived from published literature stating that there are six mice every 20 m [34], which we extrapolated out for our walkway lengths.

### 2.3. Human Risk Data

Human risk data were gathered from observations that took place on each high-use walkway. We counted the individuals who used each high-traffic walkway and categorized the risk events into behavioral risk (i.e., sitting on the grass) or clothing risk (i.e., skin exposure) on three different days, including one weekend day, for 11 hours per day. Then, we summed and scaled the data to a one-day period. The percent human risk was calculated as the number of risk events divided by the total number of observations (people) on each walkway, and is reported in the “Human Risk” column of Table 1.

### 2.4. Walkway Data

Walkway data are reported in Table 1. The Os represent organic (dirt or grass) walkways, and the NOs represent non-organic (asphalt or concrete) walkways. The length of each walkway and the average number of passersby per day per walkway are also reported. Walkways are in three different locations on the campus: the Hillside (HS) residential area, the Nature Preserve (NP), and the Susquehanna (SQ) residential area. Our primary outcome for this analysis—the simulated number of LD cases reported in the second column—was taken from a previously published study in which the number of LD cases was projected and reported for 22 different walkways and paths for the years 2008 to 2020 [33]. We used the simulated number of LD cases for the year 2014, as other data elements were gathered for the same year.

## 3. Methods

We used multinomial logistic regression [35] to investigate the effects of different factors including three types of land cover on our response variable and the number of infected ticks divided by the area of each walkway, multiplied by 1000 m^2^. Due to the convenience and availability of computer programs to apply regression analysis, this method is ubiquitous in research articles. For instance, Lorenz et al. [36] utilized logistic regression to evaluate the ability of different models in assessing the vector-borne disease risk. Moreover, Eisen and Eisen [37] introduced different techniques, including logistic regression, for preventing and controlling vector-borne disease. However, using logistic regression requires meeting its statistical assumptions to avoid any misuse or inefficient use. Therefore, we put testing this technique’s assumptions under scrutiny before developing our regression model. Some of the important assumptions of this technique are: (1) that the dependent variable should be a categorical variable, (2) observations should be independent of each other, (3) there should be no correlation among independent variables (multicollinearity), and (4) there should be a sample of at least 10 cases for each independent variable [38,39].

Using the logistic regression analysis, we try to predict the category of each walkway (as we already know the high or low risk of exposure to infected ticks in these walkways) using the independent variables available in this study.

### Data Processing

To prepare our dependent variable for this technique, we coded the risk of exposure to infected ticks in each walkway using the algorithm below:
$$\begin{array}{l} if\;\left(\frac{Number\;of\;Infected\;Ticks}{Area\;of\;walkway}\times 1000\;m^{2}\right)\\ \;<Median,\;then\;code\;it\;as\;Low\;(L)\;otherwise\;High\left(H\right) \end{array}$$

We calculated the median density of infected ticks and coded every walkway using this threshold. High (H) means that the exposure to infected ticks in a walkway is high compared to other walkways, which are Low (L).

We checked the assumptions as well by standardizing the full dataset reported in Table 1. We standardized the data by rescaling it to have a mean of zero and a standard deviation of one, because the independent and dependent variable scales differed significantly. Standardization improves the stability and parameter estimate precision in regression modeling and reduces the issue of multicollinearity, which is one of the assumptions of logistic regression analysis [40].

In addition, all the observations were collected independently; however, since walkways are connected and grouped by location, we added the location of walkways as an independent variable to our model. Moreover, we checked the multicollinearity assumption and we excluded the following variables from our analysis: density of larva, nymph, and adult ticks, and type of each walk (organic versus non-organic).

Therefore, the independent variables that were included in the final model are: three vegetation types, LD cases in 2014, number of rodents and passersby, human risk, and location of walkways.

## 4. Results and Discussion

Table 2 shows the standardized data and the final variables included in the model. The negative values in the tables are a result of standardization of the data to have a mean of zero and a standard deviation of one.

### Logistic Regression Model

We developed the first model by including all the variables in Table 2 and selecting the indicators through forward stepwise selection. However, due to the small sample size, we had a complete separation problem: the model could perfectly predict the dependent variable (this should not be the case most of the times). One common solution to this problem is to select a subset of variables to include in the model. After developing different logistic regression models, the model for which we did not get this issue included the following variables: percentage of canopy, understory, and ground cover, human risk, number of rodents and passersby.

Again, we followed the forward stepwise selection of these independent variables, and the final model reached an Akaike’s Information Criterion (AIC) of 19.06 and a Bayesian Information Criterion (BIC) of 23.43. The lower these values are, the better model we have. Moreover, the model was significant (p-value < 0.001), and the Pseudo R-square values of the model were 0.78 (Negelkerke method) and 0.63 (McFadden method). The regression equation is shown below, where “P” is the probability of being in high (H) risk walkways.$$\begin{array}{ll} Logit\left(Y\right)= & Natural\;Log\left(odds\right)=\ln\left(\frac{P}{1-P}\right)\\ & \;=2.391+\left(5.904\times \%\;of\;Understory\right)+\left(5.388\times \%\;of\;Human\;Risk\right)\\ & \;-\left(5.108\times Number\;of\;Rodents\right) \end{array}$$

The final regression model has 90.9% accuracy in classification by including the percentage of understory, human risk, and number of rodents as indicators with all having a significant contribution to the model fit, with P-values of 0.009, 0.000, and 0.001, respectively. Table 3 shows the logistic regression model results for significant indicators. The reference category in the table is the low (L) risk walkways; positive coefficients represent the tendency of walkways with a higher value of that indicator to be identified as high-risk walkways.

Walkways with a higher percentage of understory tend to fall in the “H” category (i.e., higher risk of exposure to infected ticks). Walkways with a higher percentage of human risk are associated with a lower identification of “L” class, and tend to be identified as “H” class as well (i.e., lower risk of exposure to infected ticks); this means that people tend to show more risky behaviors in high-risk walkways. These walkways are mostly in their residential area (HS) where people do not expect to be exposed to infected ticks. Therefore, awareness-based interventions in these residential areas might decrease the risk of exposure to LD.

In addition, walkways with a higher number of rodents tend to be identified as “L” class. It may be that a higher number of rodents at a walkway is associated with a lower risk of exposure, because at the nymph and adult stages (the stages that are associated with the highest risk for humans), the rodents are more easily able to groom off the ticks and maybe eat them, and thus decrease the number of ticks at a walkway. The other plausible scenario might be that passersby usually avoid those paths with higher rodents. Moreover, the availability of more natural hosts (i.e., rodents) for ticks to feed on may decrease the number of questing ticks looking for other hosts.

According to the logistic regression model, the influence of understory vegetation is significant on the risk of exposure to infected ticks. There likely will be a higher risk of exposure from walkways with understory (1–5 m) than with ground cover (less than 1 m). This is because when tick quest, seeking a blood meal as nymphs and adults, they do so by climbing onto vegetation to attach to warm-blooded mammals passing by, and much of the ground cover at the walkways we evaluated was cut lawn just an inch or two high.

The results of the current regression analysis match the results of our 2018 survey as well, in which we gathered ticks in household backyards in nearby neighborhoods. According to our results, only 9.7% of ticks are found in neighborhood residential backyard lawns (ground cover), which may support why the ground cover was not associated with the identification of high-risk walkways [41] (unpublished data).

Overall, the significant indicators of the logistic regression model are the percentage of understory, human risk, and number of rodents. All three variables are important in predicting the risk of exposure to LD. Deer are found in Hillside and Susquehanna on a regular basis around and on the walkways (whether organic or non-organic) and provide for the dispersal of ticks in the area and act as a primary breeding site for adult ticks in the fall [42]. It is clear that nymphal ticks were not collected during the summer months in the study area during the 2013–2014 season upon which the model is based. Had it been then, it is likely that nymphal ticks would have had an impact on the model. Certainly, the number of passersby played less of a role than on non-organic walkways because there was an order of magnitude fewer people traversing these earthen walkways than the paved or concrete walkways.

## 5. Conclusions

In this paper, we developed a logistic regression model on our multifaceted, unique dataset to investigate the influence of different variables such as vegetation, human risk, tick density, and rodent on the risk of exposure to infected ticks in 22 different walkways. The intersection of ecology and systems science in this research project provided the opportunity to our diverse team members, including experts in modeling, data analysis, ecology, and human population biology to identify the most influential factors associated with infected ticks and to inform urban planners and policymakers about these factors.

## 6. Limitations

For an ecological study, accessing comprehensive, big data is almost impossible. The small sample size in our study (22 observations) has likely impacted the results of our regression analysis. However, this is a unique dataset that has revealed valuable information. Furthermore, we recommend collecting more vegetation data near organic walkways to explore the influence of the percentage of canopy cover, understory cover, and ground cover in these walkways, as our data was limited and might have affected the results.

In addition, we have hypothesized that walking along walkways with areas of skin exposed may increase the risks of tick contact and LD. In subsequent studies, this hypothesis regarding LD risks can be tested and verified. For instance, people could be asked to do tick checks after they have walked along walkways or spent time sitting on the grass near a walkway, which might verify that these are risky behaviors.

We believe our analyses are useful to inform public health policy for high-use areas and walkways in built environments.

## 7. Software and/or Data Availability Section

We have provided all the data used for this research paper in the Table 1, Table 2 and Table 3. We used RStudio software for our analysis.

RStudio Team (2018). RStudio: Integrated Development for R. RStudio, Inc., Boston, MA URL http://www.rstudio.com/.

## 8. Human Subject Research

The exempt protocol number is 2316-13 for the human subject research.

## Figures and Tables

**Figure 1 healthcare-07-00066-f001:**
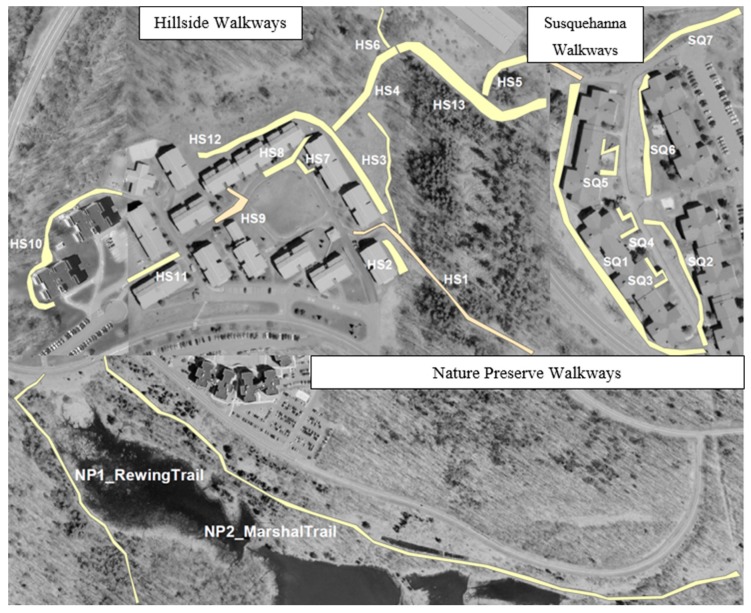
Twenty-two observed campus walkways, including two living areas (Hillside and Susquehanna) and the University Nature Preserve.

**Table 1 healthcare-07-00066-t001:** Simulated Lyme cases in 2014, walkways length and area, vegetation, tick density per 1000 m^2^, number of rodents, percentage of humans at risk, type of walkway, and average number of passersby per day by walkway for 22 high-use walkways on the Binghamton University Campus *. LD: Lyme disease.

Walkway **	Simulated LD Cases in 2014	Walkway Length (meter)	Walkway Area (meter^2^)	Canopy (%)	Understory (%)	Ground Cover (%)	Density of Larvae Ticks	Density of Nymph Ticks	Density of Adult Ticks	Density of Infected Ticks	Number of Rodents	Human Risk	Type of Walkway ***	Number of Passersby
HS1	0.0023	251.2	7787.2	50	10	84.75	0.0	0	10.7	7.705	75	0.865	NO	1690
HS2	6.2567	51.5	103	0	5	45	0.0	0	126.2	87.379	15	1	O	208
HS3	1.6712	111.9	2014.2	26	5	100	0.0	0	11.4	6.454	34	0.991	O	50
HS4	3.9596	73.5	441	40	50	67.5	0.0	0	4.5	2.268	22	0.992	O	119
HS5	4.1989	44.2	699.8	27.5	6.25	70	0.0	0	1.4	1.429	13	1	O	123
HS6	1.1354	44.8	627.2	55	87.5	15	4.8	41.5	3.2	14.349	13	1	O	26
HS7	0.0060	24.7	148.2	35	0	100	0	0	6.7	6.748	7	0.678	NO	649
HS8	0.0007	17.7	106.2	30	0	100	0	0	0	0	5	0.489	NO	574
HS9	0.0005	42.7	256.2	20	0	100	0	0	0	0	13	0.400	NO	800
HS10	0.00006	148.7	2676.6	7.5	12.5	55	0	0	2.2	0.374	45	0.562	O	126
HS11	0.00007	60.35	724.2	14.75	0	30	0	0	0	0	18	0.757	NO	573
HS12	0.00009	215.5	1939.5	2.5	5	97.5	0	0	2.1	0.516	65	0.983	O	118
HS13	0.00006	154.5	4480.17	47.5	35	53.75	0	0.7	19.2	12.5	46	1	O	60
NP-MT	0	1134.8	26100.4	51.5	52.5	75	0	0	0.3	0.115	340	1	O	639
NP-RW	0.0004	548.6	3291.6	95	10	56	0	0	0.6	0.304	165	0.999	O	168
SQ1	0.00003	296.08	2072.56	20	15	100	0	2.9	0	0.482	89	0.824	O	25
SQ2	0.0002	120.31	721.86	5	6.75	97.5	0	0	2.8	0	36	0.646	NO	213
SQ3	0.0006	34.58	69.16	30	5	100	0	0	0	0	10	0.729	NO	391
SQ4	0.0007	41.47	82.94	5	0	95	0	0	0	0	12	0.697	NO	577
SQ5	0.0007	53.18	106.36	20	0	80	0	0	0	0	16	0.755	NO	588
SQ6	0.0003	38.5	462	7.5	17.5	80	0	0	6.5	0	12	0.527	NO	341
SQ7	1.3754	45.03	495.33	3.75	35	65	0	0	22.2	8.075	14	0.573	NO	1295

* The State University of New York at Binghamton is a mid-sized university with a 930-acre campus in Broome County, New York. ** HS = Hillside walkways; NP = Nature Preserve Walkways; SQ = Susquehanna Walkways. *** NO = Non-organic Walkway; O = Organic Walkway.

**Table 2 healthcare-07-00066-t002:** Standardized data for walkways—infected tick risk is coded to account for high (H) and low (L) risk of exposure to infected ticks in each walkway—Hillside walkways are coded as 1, Nature Preserve walkways are coded as 2, and Susquehanna walkways are coded as 3.

Walkway **	Infected Tick Risk (Dependent Variable)	Location	Simulated LD Cases in 2014	Canopy (%)	Understory (%)	Ground Cover (%)	Number of Rodents	Human Risk	Number of Passersby
**HS1**	H	1	−0.488	1.001	−0.279	0.358	0.355	0.353	2.967
**HS2**	H	1	3.131	−1.173	−0.502	−1.227	−0.446	1.026	−0.510
**HS3**	H	1	0.477	−0.042	−0.502	0.966	−0.192	0.981	−0.880
**HS4**	H	1	1.802	0.566	1.501	−0.330	−0.352	0.986	−0.718
**HS5**	H	1	1.94	0.023	−0.446	−0.230	−0.472	1.026	−0.709
**HS6**	H	1	0.167	1.219	3.169	−2.423	−0.472	1.026	−0.935
**HS7**	H	1	−0.486	0.349	−0.724	0.966	−0.552	−0.576	0.525
**HS8**	L	1	−0.489	0.131	−0.724	0.966	−0.579	−1.519	0.350
**HS9**	L	1	−0.489	−0.303	−0.724	0.966	−0.472	−1.960	0.879
**HS10**	L	1	−0.489	−0.847	−0.168	−0.828	−0.045	−1.154	−0.702
**HS11**	L	1	−0.489	−0.532	−0.724	−1.825	−0.406	−0.183	0.347
**HS12**	H	1	−0.489	−1.064	−0.502	0.866	0.221	0.940	−0.720
**HS13**	H	1	−0.489	0.892	0.833	−0.878	−0.032	1.026	−0.858
**NP-MT**	L	2	−0.489	1.066	1.612	−0.031	3.888	1.026	0.502
**NP-RW**	L	2	−0.489	2.958	−0.279	−0.788	1.555	1.019	−0.603
**SQ1**	H	3	−0.489	−0.303	−0.057	0.966	0.541	0.148	−0.939
**SQ2**	L	3	−0.489	−0.956	−0.424	0.866	−0.165	−0.737	−0.497
**SQ3**	L	3	−0.489	0.131	−0.502	0.966	−0.512	−0.323	−0.081
**SQ4**	L	3	−0.489	−0.956	−0.724	0.767	−0.486	−0.482	0.355
**SQ5**	L	3	−0.489	−0.303	−0.724	0.169	−0.432	−0.193	0.381
**SQ6**	L	3	−0.489	−0.847	0.055	0.169	−0.486	−1.329	−0.197
**SQ7**	H	3	0.306	−1.010	0.833	−0.429	−0.459	−1.098	2.042

**Table 3 healthcare-07-00066-t003:** Multinomial logistic regression results for significant indicators.

Significant Indicator (High-Risk Walkways)	Coefficient Estimates	Standard Errors	*p*-Value
**% of Understory**	5.9	3.84	0.009
**% of Human Risk**	5.388	3	0.000
**Number of Rodents**	−5.108	2.86	0.001

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
