# Peer review of "Risk Factors of Lyme Disease: An Intersection of Environmental Ecology and Systems Science"

_healthcare, 2019, doi:10.3390/healthcare7020066_

Round 1

Reviewer 1 Report

The authors have improved the article and in its present form is acceptable.

Author Response

We are glad that we could address the reviewer's concerns.

Thank you.

Reviewer 2 Report

I appreciate the authors’ attempts to address my concerns regarding their manuscript.  However, I still disagree with their approach to analyzing their data.  Primarily, I still believe LD cases (or their normalized version, LD cases/length * 100 m) is an inappropriate dependent variable.  The counts of LD cases are simulated from the model they described and used to evaluate the hypothetical effects of various interventions on LD (Sharareh et al. 2017).  I realize that paper is not the paper being reviewed here and it is already published, but there are problems that I have with that paper, which may help the authors understand why I’m having trouble with their use of simulated LD cases from that model as the dependent variable in the paper currently being reviewed.  Overall, I don’t have problems with the conceptual development of their model, they integrated important complex, interconnected components of LD transmission in the model; however, they calibrate or evaluate the accuracy of their model using historic data, the number of LD cases 2008-2016, by scaling down the number of LD cases reported to the NYSDH for Broome County, the county in which Binghamton University exists.  This was done by adjusting the number based on the size of the campus population in comparison to the county population (e.g., LD cases for BU campus = 20% increased adjusted LD cases for Broome County * (BU campus population/Broome County population)).  The number of LD cases generated by their model closely fit the scaled-down number, which suggests the model is working.  However, there is the major assumption that the scaled-down number of LD cases for the BU campus is an accurate estimate of the number of LD cases on the BU campus, but there is no evaluation of this, therefore, it puts the whole use of the model-simulated LD cases as a measurement of LD risk along walkways into question.  There are no real counts of LD cases along the walkways or on the BU campus to assess the validity of the findings they report.  Therefore, if there is no way for the investigators to get LD counts for the BU campus, such as from the University’s health center, then I would like to repeat that I think their analysis and manuscript would be more acceptable if they used a variable that they actually measured, like, density of ticks, or density of infected ticks, as their dependent variable.  In fact, density of nymphs (DON) and density of infected nymphs (DIN) are standard variables used to assess human risk of LD (they reference a number of papers that use these variables, see Eisen & Eisen, Eisen et al. and Diuk-Wasser papers), so there use of these variables would be in line with what others have done.  Again, I would be okay with them using their model to generate the simulated number of LD cases along their walkways and to discuss, but I don’t think it should be the dependent variable used in their analyses to determine which walkways have a high or low risk of generating LD cases.

If I had the authors’ dataset, this is how I would approach the analysis:

Use DON, DIN, or some other measurement of density of ticks or infected ticks  collected along walkways as the dependent variable (=predicted human LD risk) in an analysis to determine which of the physical and vegetation characteristics of the walkways were significantly associated with that tick measurement.

Then, classify the walkways as high or low risk based on densities of ticks/infected ticks and compare the human use and risky use variables that were measured between these walkways.

Plug measured values of tick and human parameters into simulation model to generate simulated numbers of LD cases along walkways and discuss which walkways are expected to be associated with more LD cases because of the characteristics of the walkways and human use of walkways.  The simulation of LD cases in this way is essentially a model-generated hypothesis that can be tested by either comparing to numbers of actual, real LD cases that were contracted along walkways, or that can be supported by doing the experiments to see if people pick up ticks walking along or spending time along the walkways.

Essentially, you don't have to try to do everything in just one regression analysis.  You can tell a story, based on the results of more than one statistical analysis, done in an attempt to understand inter-related parts to LD transmission and human LD risk.

Other comments:

Authors still did not state whether or not ticks were surveyed along all walkways during the same seasons (and this is not provided in Sharareh et al. 2017).  That is, were all walkways sampled in the spring/summer to assess for larvae and nymphs and were all walkways sampled in fall for adults?  If not, or if the same drag effort was not expended along each walkway, tick numbers should be adjusted by effort.

It isn’t clear why estimates for the densities of P. leucopus along walkways used here were derived from literature rather than the trapping that they did (reported in Sharareh et al. 2017).

Walkway labels in Figure 1 are still too small/difficult to see in the figure.

Author Response

We thank the editor and anonymous reviewers for their useful comments. We appreciate the opportunity to address the comments by submitting a revision. We have considered all comments precisely, and we believe the revisions address these comments and have greatly improved the paper. In this revision report, we have addressed each reviewer’s concerns by explaining in detail the responses (In italic) to their questions/concerns (numbered).

Reviewer 2 

1. Use DON, DIN, or some other measurement of density of ticks or infected ticks  collected along walkways as the dependent variable (=predicted human LD risk) in an analysis to determine which of the physical and vegetation characteristics of the walkways were significantly associated with that tick measurement. Then, classify the walkways as high or low risk based on densities of ticks/infected ticks and compare the human use and risky use variables that were measured between these walkways.

We appreciate the comments. Based on the available data, discussion with our research team, and reviewer’s comment, we decided to use the density of infected ticks (nymphs and adults) to classify walkways into “high” or “Low” risk walkways using the median of “density of infected ticks” in 22 walkways.

if (density of infected ticks per 1000 m2) < median, then code it as Low (L) otherwise High (H)

Hence, our dependent variable is a categorical variable and Using the logistic regression analysis, we try to predict the category of each walkway (as we already know the high or low risk of exposure to infected ticks in these walkways) using the independent variables available in this study.

Also, we kept the simulated LD cases in 2014 and considered it as an independent variable. However, it was not used in our final model due to the assumptions of the logistic regression model.

2. Plug measured values of tick and human parameters into simulation model to generate simulated numbers of LD cases along walkways and discuss which walkways are expected to be associated with more LD cases because of the characteristics of the walkways and human use of walkways.  The simulation of LD cases in this way is essentially a model-generated hypothesis that can be tested by either comparing to numbers of actual, real LD cases that were contracted along walkways, or that can be supported by doing the experiments to see if people pick up ticks walking along or spending time along the walkways.

This is a great idea and we really appreciate that. However and for the sake of this paper, we skip this step, and we will consider it in our future research endeavors.

3. Authors still did not state whether or not ticks were surveyed along all walkways during the same seasons (and this is not provided in Sharareh et al. 2017).  That is, were all walkways sampled in the spring/summer to assess for larvae and nymphs and were all walkways sampled in fall for adults?  If not, or if the same drag effort was not expended along each walkway, tick numbers should be adjusted by effort.

All walkways were surveyed in fall and spring, and each one between 1 and 4 times (This statement has been added to the “Tick-Rodent Data” section of the manuscript).

In terms of the reviewer's comment on adjusting the tick numbers by effort, we would have to disagree. We are doing density per 1000 square meters; hence, it is all relative. Our unit of measure is not hours dragged, or times dragged, it is area dragged and we have calculated it out so each walkway is equal-- per 1000 square meters.

4. It isn’t clear why estimates for the densities of P. leucopus along walkways used here were derived from literature rather than the trapping that they did (reported in Sharareh et al. 2017).

Although we collected mice throughout the campus, this data was very limited; hence, we decided to estimate the mice population using the literature. As it was explained in Sharareh et al (2017): “The number of mice in the area was estimated based on the reproduction and longevity information from Aguilar [44].

This statement has been added to the “Tick-Rodent Data” section of the manuscript.

5. Walkway labels in Figure 1 are still too small/difficult to see in the figure.

The figure is updated.

Round 2

Reviewer 2 Report

I thank the authors for taking the time to redo the analyses with my suggestions.  Changing the dependent variable in the analysis from number of simulated LD cases to the number of infected tick, normalized by length of walkway, and then coded into high or low categories for logistic regression, is acceptable.

I think it would be helpful to have a table of the significant independent factors in the final model, listing the coefficient estimates, standard errors, and p-values for each factor listed, but this is not completely necessary.

The accuracy of classification of a model is usually evaluated using a new dataset, or a subset of data that were excluded from model development, not using data used to develop model, as was done for Table 3 in the paper.  Therefore, I don't think this Table or references to the accuracy of model should be included.  The statistical method will find the model with the best fit of the data used, therefore, you should have high classification accuracy with the data you used to fit the model.

An additional explanation for why higher P. leucopus densities may be associated with lower tick risk, may be that with ticks have more natural hosts to feed on in such areas such that there may be fewer questing ticks looking for a host.

Some editing to improve English or correct grammar is needed.  In particular, description of relationship between tick risk and Human Risk factor in model on lines 268-273 needs improvement.  Not clearly stated here. 

Author Response

We thank the editor and anonymous reviewers for their useful comments. We have considered all comments precisely, and we believe the revisions address these comments and have greatly improved the paper. In this revision report, we have addressed each reviewer’s concerns by explaining in detail the responses (In italic) to their questions/concerns (numbered).

Reviewer 2 

1. I think it would be helpful to have a table of the significant independent factors in the final model, listing the coefficient estimates, standard errors, and p-values for each factor listed, but this is not completely necessary.

2. The accuracy of classification of a model is usually evaluated using a new dataset, or a subset of data that were excluded from model development, not using data used to develop model, as was done for Table 3 in the paper.  Therefore, I don't think this Table or references to the accuracy of model should be included.  The statistical method will find the model with the best fit of the data used, therefore, you should have high classification accuracy with the data you used to fit the model.

We removed Table 3 and added the suggested Table from previous comments. More information were also added above the table.

3. An additional explanation for why higher P. leucopus densities may be associated with lower tick risk, may be that with ticks have more natural hosts to feed on in such areas such that there may be fewer questing ticks looking for a host.

Thank you for the suggestions. We added this to the discussion section.

4. Some editing to improve English or correct grammar is needed.  In particular, description of relationship between tick risk and Human Risk factor in model on lines 268-273 needs improvement.  Not clearly stated here. 

This section is revised and the whole manuscript were proofread again.

This manuscript is a resubmission of an earlier submission. The following is a list of the peer review reports and author responses from that submission.

Round 1

Reviewer 1 Report

Sharareh et al. report results from a study that examined potential Lyme disease (LD) risk along walkways on a university campus in upstate New York.  Factors evaluated as determinants of LD risk were related to the material composition of the walkways (organic versus non-organic), characteristics of the vegetation boarding walkways, abundance of ticks and infected ticks collected, estimates of Peromyscus leucopus abundance, as well as measures related to human use along the walkways.

 I appreciate the researchers’ efforts to examine LD risk in an environment that humans use for normal, everyday activities, as well as their efforts to examine the combination of environmental and human-related factors as possible determinants of LD risk.  There actually are very few studies that have attempted to do these things, even though understanding where and how humans get infected is critically important for identifying ways to prevent infections.  However, I view their analysis as being overly complicated, and perhaps, done incorrectly or inappropriately in some respects.

The study first suffers from a small sample size of 22 walkways, which really is difficult to avoid for an ecological study, such as the one done by the researchers.  There simply are limitations related to the logistics of conducting drags for ticks, or for making observations of humans on the walkways, that put constraints on a study’s sample size—these activities are time consuming and expensive if people are being paid to do them.  The input or independent variables that were measured and evaluated in their analysis also are complex and difficult to measure accurately or precisely, such that one or two measurements collected probably don’t really reflect the “true” value of the variables at the walkways.  

The small sample size and difficult nature of measuring the independent variables then, puts limits on what types of analyses can be done and how results should be interpreted.  The researchers attempt to use multiple regression analysis, specifically applying the least squares method, and to treat all variables as continuous.  They evaluate the data in terms of meeting the statistical assumptions, primarily related to the assumptions of a linear model, and modify the data to try to get them to meet the assumptions.  For instance, they standardize all variables by rescaling them to have a mean of 0 and standard deviation of 1, because the variables fail to show a linear relationship to their dependent variable.  When that failed to linearize relationships, they decided to do separate analyses for the organic and non-organic walkways, and when that failed, they removed outliers.  Such that they end up further reducing the sample sizes, and power, of their analyses.  Moreover, they use significance of the Pearson correlation statistic as a measure of linearity, which probably isn’t the best way to assess linearity anyway—I would suggest looking at scatter plots to visualize the relationship between 2 variables, or more appropriately for multiple regression, visualize a plot of the residuals against the fitted values to make sure there isn’t any curvature to the points.  The Pearson correlation coefficient r quantifies the strength and direction, assuming a linear relationship, but visualization of the plots will tell you if the relationship is actually linear (or close enough, it doesn’t have to be perfect).  Also, looking at the data in Table 1, I would suggest not trying to treat all variables as continuous.  The densities of larvae and nymph ticks, in particular, were zeros along most walkways such that these probably should be converted to categorical (e.g., presence, absence).  Removing outliers also is generally not encouraged, especially when you have such few data points to begin with.  Forcing data to fit a particular model is not the way to do analyses, rather data should be analyzed using the most appropriate methods for the data! 

Other problems with the analytic approach:  the assumption of independence among data points is probably violated—walkways are connected, grouped by location; therefore, location (e.g., HS, NP, SQ) should be included as an independent variable to evaluate that effect. I also believe the dependent or outcome variable used in the analysis is inappropriate.  It’s completely “made up,” based on the simulated number of Lyme disease cases from a model the researchers described in a previous paper (Sharareh et al. 2017).  And, assessment of variables included in the models and the overall models themselves weren’t done as I would expect.  Typically, independent variables are assessed in terms of their p-values in the models and removed or added as appropriate (no table or report of these p-values is included).  The r2 value of the model is a measure of how closely the linear regression model fits the data, not so much the “strength” of the relationship between independent and dependent variables.  If the relationships between independent and dependent variables aren’t linear, the model won’t fit very well and the r2 value will be lower. Other models (not linear) may be more appropriate.

I would recommend that researchers take a simpler, more descriptive approach to presenting their data.  They do have a unique dataset, investigating the potential LD risk associated with walkways, which gets at the basic question that we still don’t have a good answer to:  “Where do people get infected with LD?”  I think, if they refocused their analysis to describing the potential risk people walking along the walkways may have in contacting ticks, that would be make for a better paper.  I would use densities of ticks as the outcome/dependent variable, and evaluate which variables (locations, type of walkway, vegetation) were associated with finding more questing ticks.  T-tests or ANOVAs, or nonparametric equivalents, could be used here.  Then, a description of the human use of the various walkways with different risks could be presented (e,g., simple means or medians and some measure of variation for number of passersby and proportion of passersby doing various activities).  It would be okay to report the simulated LD cases for the walkways and discuss, but I don’t think that should be the outcome variable used in analyses. 

Essentially, the researchers have generated data that allows them to hypothesize about  how LD risk may be associated with walking along organic walkways, or how walking along walkways with areas of skin exposed may increase the risks of tick contact and LD, but an evaluation of those risks have not really been done yet.  Actually, the hypotheses regarding risks could be tested in subsequent studies.  For instance, people could be asked to do tick checks after they’ve walked along walkways or spent time sitting on grass near a walkway (or the researchers could do these activities themselves), to see if these really are risky behaviors. 

Other problems with the paper.  More information about how tick densities were estimated would be appreciated.  For instance, density of questing ticks varies tremendously by season, day, and hour of day, depending on the weather, vegetation, and all hosts available (not only P. leucopus).  It isn’t clear from the researchers’ descriptions here or in their previous paper (Sharareh et al. 2017) how the densities of larvae, nymphs, and adults were actually estimated.  How frequently were walkways sampled for ticks? Were they all sampled in the same seasons? 

Variables included in Table 1 should have units of measurement included.  For instance, density of ticks should be reported as number per meter or meter2.  A better description of how Human Risk was calculated is needed, as well as number of rodents.

In Figure 1, the walkway labels should be enlarged.  As is, they aren’t visible.

In the first paragraph of the Introduction, lines 42-44, Post-treatment LD syndrome specifically refers to people that are treated, but continue to have symptoms.  This is separate from the progression of LD to more complex manifestations that occurs in people that are left untreated.  Please correct.

Author Response

First of all, we would like to thank the editor and anonymous reviewers for their useful comments and constructive critique. We appreciate the opportunity to address the comments by submitting a revision. We have considered all comments precisely, and we believe the revisions address these comments and have greatly improved the paper. In this revision report, we have addressed each reviewer’s concerns by explaining in detail the responses (In italic) to their questions/concerns (numbered).

1. The study first suffers from a small sample size of 22 walkways, which really is difficult to avoid for an ecological study, such as the one done by the researchers.  There simply are limitations related to the logistics of conducting drags for ticks, or for making observations of humans on the walkways, that put constraints on a study’s sample size—these activities are time consuming and expensive if people are being paid to do them.  The input or independent variables that were measured and evaluated in their analysis also are complex and difficult to measure accurately or precisely, such that one or two measurements collected probably don’t really reflect the “true” value of the variables at the walkways.  

We really appreciate your understanding of limited availability of such a data.

For some of the independent variables, including human risk data and number of passersby, we agree with you that we had to consider simplification techniques and apply assumptions to derive the data. However, to the best of our knowledge, this is the best kind of dataset available to researchers. Regarding other independent variables, our anthropologists considered valid, comprehensive techniques to gather data and we believe that the data are reflecting the reality.

2. The small sample size and difficult nature of measuring the independent variables then, puts limits on what types of analyses can be done and how results should be interpreted.  The researchers attempt to use multiple regression analysis, specifically applying the least squares method, and to treat all variables as continuous. They evaluate the data in terms of meeting the statistical assumptions, primarily related to the assumptions of a linear model, and modify the data to try to get them to meet the assumptions. For instance, they standardize all variables by rescaling them to have a mean of 0 and standard deviation of 1, because the variables fail to show a linear relationship to their dependent variable. When that failed to linearize relationships, they decided to do separate analyses for the organic and non-organic walkways, and when that failed, they removed outliers. Such that they end up further reducing the sample sizes, and power, of their analyses. Moreover, they use significance of the Pearson correlation statistic as a measure of linearity, which probably isn’t the best way to assess linearity anyway—I would suggest looking at scatter plots to visualize the relationship between 2 variables, or more appropriately for multiple regression, visualize a plot of the residuals against the fitted values to make sure there isn’t any curvature to the points. The Pearson correlation coefficient r quantifies the strength and direction, assuming a linear relationship, but visualization of the plots will tell you if the relationship is actually linear (or close enough, it doesn’t have to be perfect). Also, looking at the data in Table 1, I would suggest not trying to treat all variables as continuous. The densities of larvae and nymph ticks, in particular, were zeros along most walkways such that these probably should be converted to categorical (e.g., presence, absence).  Removing outliers also is generally not encouraged, especially when you have such few data points to begin with.  Forcing data to fit a particular model is not the way to do analyses, rather data should be analyzed using the most appropriate methods for the data! 

Thank you for the suggestions.

We considered another method—Logistic Regression—instead of multiple regression analysis. See the “methods” section for our justifications.

We did not remove the outliers in this revision.

All our 22 observations are considered together for the logistic regression model and we did not divide walkways into organic and non-organic ones.

We considered the density of infected nymphs and larva as categorical variables; density of larva was correlated with other independent variables and were not included in the model.

3. Other problems with the analytic approach:  the assumption of independence among data points is probably violated—walkways are connected, grouped by location; therefore, location (e.g., HS, NP, SQ) should be included as an independent variable to evaluate that effect. I also believe the dependent or outcome variable used in the analysis is inappropriate.  It’s completely “made up,” based on the simulated number of Lyme disease cases from a model the researchers described in a previous paper (Sharareh et al. 2017).  And, assessment of variables included in the models and the overall models themselves weren’t done as I would expect.  Typically, independent variables are assessed in terms of their p-values in the models and removed or added as appropriate (no table or report of these p-values is included).  The r2 value of the model is a measure of how closely the linear regression model fits the data, not so much the “strength” of the relationship between independent and dependent variables.  If the relationships between independent and dependent variables aren’t linear, the model won’t fit very well and the r2 value will be lower. Other models (not linear) may be more appropriate.

We included the location as an independent variable, however, it was not included in the final logistic regression model.

Regarding the LD cases, there is no historical data on number of LD cases for different walkways along with their environmental and ecological factors, to be used instead for our regression model. On the other hand, in the reference article [33], number of LD cases for each walkway is simulated based on a simulation model that has been validated using rigorous methods explained in reference article [33]. Since we had data for the ecological and environmental factors for different walkways, the best approach was to get the simulated LD cases for those walkways as well, to be able to produce a comprehensive model that captures the influence of ecological, environmental, and human behavioral risk on LD cases.

Final significant indicators have been reported with their p-values.

The validity of the final model has been tested using different indices such as AIC, BIC, Pseudo R-Square, and P-value of the model.

4. I would recommend that researchers take a simpler, more descriptive approach to presenting their data.  They do have a unique dataset, investigating the potential LD risk associated with walkways, which gets at the basic question that we still don’t have a good answer to:  “Where do people get infected with LD?”  I think, if they refocused their analysis to describing the potential risk people walking along the walkways may have in contacting ticks, that would be make for a better paper.  I would use densities of ticks as the outcome/dependent variable, and evaluate which variables (locations, type of walkway, vegetation) were associated with finding more questing ticks.  T-tests or ANOVAs, or nonparametric equivalents, could be used here.  Then, a description of the human use of the various walkways with different risks could be presented (e,g., simple means or medians and some measure of variation for number of passersby and proportion of passersby doing various activities).  It would be okay to report the simulated LD cases for the walkways and discuss, but I don’t think that should be the outcome variable used in analyses. 

Our dependent variable in this revision is (LD Cases/Length of each walkway)*100 meter.

5. Essentially, the researchers have generated data that allows them to hypothesize about  how LD risk may be associated with walking along organic walkways, or how walking along walkways with areas of skin exposed may increase the risks of tick contact and LD, but an evaluation of those risks have not really been done yet.  Actually, the hypotheses regarding risks could be tested in subsequent studies.  For instance, people could be asked to do tick checks after they’ve walked along walkways or spent time sitting on grass near a walkway (or the researchers could do these activities themselves), to see if these really are risky behaviors. 

We appreciate the reviewer’s comments and have added the following statement to the ‘Limitations’ section: “In addition, we have hypothesized that LD risk may be associated with walking along organic walkways, and walking along walkways with areas of skin exposed may increase the risks of tick contact and LD. In subsequent studies, these hypotheses regarding LD risks can be tested and verified. For instance, people could be asked to do tick checks after they have walked along walkways or spent time sitting on grass near a walkway, to verify that these are risky behaviors.”

6. Other problems with the paper.  More information about how tick densities were estimated would be appreciated.  For instance, density of questing ticks varies tremendously by season, day, and hour of day, depending on the weather, vegetation, and all hosts available (not only P. leucopus).  It isn’t clear from the researchers’ descriptions here or in their previous paper (Sharareh et al. 2017) how the densities of larvae, nymphs, and adults were actually estimated.  How frequently were walkways sampled for ticks? Were they all sampled in the same seasons? 

The tick data module was obtained from our reference article and detailed information are available in that paper, which explain the details extensively. In order to clarify the data in current paper, we added following statement under section 2.2:

“Black legged ticks (Ixodes scapularis), the primary vector, were collected within 3m on each side of all 22 walkways for a total area of 71 534m2 (7.1 ha) during two consecutive years, 2013 and 2014. Ticks were collected using the traditional dragging method except for winter months. Ticks were transported from the cloths to the laboratory for pathogen analysis. Tick densities could be calculated using the number of ticks collected and the area dragged, which is discussed in detail by Sharareh, et al. [33].”

7. Variables included in Table 1 should have units of measurement included.  For instance, density of ticks should be reported as number per meter or meter2.  A better description of how Human Risk was calculated is needed, as well as number of rodents.

We added the “density per 1,000 meter2” to the caption of Table 1. For Human Risk and Rodents calculations, we provided further details under section 2.3 Human Data section. For more information, please refer to Sharareh, et al. [33].

8. In Figure 1, the walkway labels should be enlarged.  As is, they aren’t visible.

Figure is updated.

9. In the first paragraph of the Introduction, lines 42-44, Post-treatment LD syndrome specifically refers to people that are treated, but continue to have symptoms.  This is separate from the progression of LD to more complex manifestations that occurs in people that are left untreated.  Please correct.

We corrected this sentence to "Initial symptoms of infection are often characterized by a rash, severe fatigue, joint pain, and neurological symptoms [12]. If left untreated, symptoms may progress and present as complex multi-system manifestations [12, 13].

Reviewer 2 Report

The article by Sharareh et al., entitled “Risk Factors of Lyme Disease: An Intersection of Environmental Ecology and System Science” uses a regression analysis to elucidate the effect of different conditions on the number of LD cases.

Major Issues

1. It is unclear how the number of LD cases was calculated. The authors used modelling from a previous publication, which is a major limitation since there is not actual data on Lyme disease cases in this study. Thus, the relevance of the regression analysis is extremely limited.

2. The standardization of the data may allow the authors to perform the regression analysis, however it does not make sense to have negative numbers for rodents, density of ticks, etc…

3. The study does not take into consideration seasonality, which is a major factor in Lyme disease cases.

4. It is unclear why the authors included density of larvae ticks since they do not transmit Lyme disease.

5. It is unclear where the density of infected nymphs and adults is coming from; it seems that there was not testing involved in this study.

6. The density of mice is extrapolated from a reference. One would think that areas with different vegetation may influence the number of mice. Importantly, immature ticks feed on small mammals and if mice are not available, they may feed on other hosts which will affect the percentage of infected ticks in the area since not all potential I. scapularis hosts are equally competent in transmitting Lyme disease.

Author Response

First of all, we would like to thank the editor and anonymous reviewers for their useful comments and constructive critique. We appreciate the opportunity to address the comments by submitting a revision. We have considered all comments precisely, and we believe the revisions address these comments and have greatly improved the paper. In this revision report, we have addressed each reviewer’s concerns by explaining in detail the responses (In italic) to their questions/concerns (numbered).

1. It is unclear how the number of LD cases was calculated. The authors used modelling from a previous publication, which is a major limitation since there is not actual data on Lyme disease cases in this study. Thus, the relevance of the regression analysis is extremely limited.

As mentioned in the article, this data was taken from our reference article. They had simulated number of LD cases for each walkway that is generated from their validated simulation model. Unfortunately, there is no historical data on number of LD cases for different walkways along with their environmental and ecological factors. Since we had data for the ecological and environmental factors for different walkways, the best approach was to get the simulated LD cases for those walkways as well. Otherwise, we could not use the unique ecological data that we had.

2. The standardization of the data may allow the authors to perform the regression analysis, however it does not make sense to have negative numbers for rodents, density of ticks, etc…

There are different methods for standardizing the data. In the approach that we considered (i.e. mean of zero and standard deviation of one) all the independent variables have a mean of zero, and by using the standard deviation of one, some of them get negative values and some positive, however, it does not mean that we have negative number of rodents or ticks. It is just a data analysis technique to prepare the data for analysis and the nature of data will not be altered (i.e. smaller numbers in each variable will have negative values after standardization).

3. The study does not take into consideration seasonality, which is a major factor in Lyme disease cases.

We appreciate this comment. Seasonality was considered in our reference article where simulation modeling was used to replicate the historical data of LD cases. Therefore, we have considered this indirectly. However, seasonality and considering that as an independent variable is a complex matter and requires comprehensive data that is not available to us.

4. It is unclear why the authors included density of larvae ticks since they do not transmit Lyme disease.

Although Larvae ticks do not transfer Lyme disease, but they are an important part of the tick’s life cycle and some studies have proposed controlling their population, as they will finally molt into nymphs and adults. Therefore, considering their density allows us to have a broader analysis. However, for the resubmission, we changed the method of analysis, and this variable had correlation with other independent variables and we decided to avoid including it in the model to meet the multicollinearity assumption.

5. It is unclear where the density of infected nymphs and adults is coming from; it seems that there was not testing involved in this study.

Some of the data used in this study were taken from our reference article. We specifically mention these points in our revision, first paragraph of “Data” section, as follows:

“The data includes four different modules. Vegetation data were gathered specifically for the purpose of this research article from 22 different walkways at the mid-sized university campus mentioned above; data for other modules were obtained from a previous published article [33] available for all those 22 walkways. Sharareh et al [33] used systems science techniques including data analysis and simulation to develop a model in order to propose policies capable of reducing LD cases. Below, we explain each module in detail.”

6. The density of mice is extrapolated from a reference. One would think that areas with different vegetation may influence the number of mice. Importantly, immature ticks feed on small mammals and if mice are not available, they may feed on other hosts which will affect the percentage of infected ticks in the area since not all potential I. scapularis hosts are equally competent in transmitting Lyme disease.

That is correct and we appreciate this comment. We have checked this fact in our data, and there is no correlation between mice and vegetation types. Therefore, we included both of them in our regression model.

Reviewer 3 Report

The narrative was heavy with terminology and statistics considered commonplace in the field of ecology, but perhaps a little too heavy for a journal such as Healthcare. This was mostly just an issue of how hard the reader has to work to understand the message.

Specific Comments

 Line 220 Remove “were”

Author Response

First of all, we would like to thank the editor and anonymous reviewers for their useful comments and constructive critique. We appreciate the opportunity to address the comments by submitting a revision. We have considered all comments precisely, and we believe the revisions address these comments and have greatly improved the paper. In this revision report, we have addressed each reviewer’s concerns by explaining in detail the responses (In italic) to their questions/concerns (numbered).

1. Line 220 Remove “were”

We deleted it. Thank you.